# Volabolomic Fingerprinting for Post-Mortem Interval Estimation: A Novel Physiological Approach

**DOI:** 10.3390/biom14030286

**Published:** 2024-02-28

**Authors:** Andrea Mazzatenta, Tiziana Pietrangelo, Roberto Demontis, Cristian D’Ovidio

**Affiliations:** 1Neuroscience, Imaging and Clinical Science Department, “G. d’Annunzio” University of Chieti-Pescara, 66100 Chieti, Italy; tiziana.pietrangelo@unich.it; 2Dipartimento di Scienze Mediche e Sanità Pubblica, Università degli Studi di Cagliari and Azienda Ospedaliero-Universitaria di Cagliari, 09123 Cagliari, Italy; demrob@unica.it; 3Medicine and Aging Sciences Department, “G. d’Annunzio” University of Chieti-Pescara, 66100 Chieti, Italy; cristian.dovidio@unich.it

**Keywords:** PMI, post-mortem interval estimation, volabolomics, volatile metabolite biomarkers, e-nose, forensic science, human decomposition, forensic metabolomics, skeletal muscle

## Abstract

Death is a multifaceted process wherein each individual cell and tissue has a metabolic homeostasis and a time of functional cessation defined by the dying process as well as by intrinsic and extrinsic factors. Decomposition is physiologically associated with the release of different types of volatile organic compounds (VOCs), and these form volaboloma mortis. The main purpose of this study was to record the volabolomic fingerprint produced by volatile molecules during the physiological decomposition process of human tissue and muscle cells. The volatile chemical signature has important implications for an open issue in forensics and pathology, namely the estimation of the postmortem interval (PMI), which decreases in accuracy with the passage of time. Volatile metabolites emitted from human tissues and muscle cells at 0, 24, 48, and 72 h were recorded in real time with an electronic nose sensor device. The key findings were the continuous sampling of VOCs emitted from tissues and cells. These showed a common behavior as time progressed; particularly, after 48 h the distributions became dispersed, and after 72 h they became more variable. Volabolomic fingerprinting associated with time progression relevant to the study of PMIs was reconstructed. Additionally, there may be broader applications, such as in dog training procedures for detecting human remains, and perhaps even for studying scavenger and insect attractants.

## 1. Introduction

Postmortem interval (PMI) refers to the estimation of time of death, and is a pivotal element of forensic medicine and pathology. As time since death progresses, the PMI estimate decreases in accuracy. Several methods have been developed to estimate the PMI of cadavers in different stages of decay. However, accurate estimation can be challenging, especially when dealing with corpses in advanced stages of decomposition [1,2,3,4,5]. Several postmortem (PM) markers can be used to ascertain the time elapsed since death in the initial PM period. In the first 48–72 h after death, the body undergoes several postmortem changes used by pathologists and referred to as rigor mortis, indicating increased rigidity associated with muscle stiffness; algor mortis, progressive cooling; and livor mortis, the pink-purplish discoloration in fair-skinned individuals caused by the lack of blood circulation and sedimentation of blood in lower areas [1,6,7]. Additional laboratory systems for PMI estimation involving supravital tissue reactions are, for example, ocular analysis, including potassium and hypoxanthine in the vitreous, pharmacological excitability of the iris, and electrical stimulation of the eyelid muscle, as well as analysis of biological clock genes, etc. Like other methods, these lose value and accuracy as decay progresses and organs and other soft tissues liquefy [1,8].

Returning to the physiological aspects of death, the cessation of blood circulation and tissue oxygenation progressively produces cellular hypoxia and anoxia. This leads to the swelling of cells, the subsequent rupture of cell membranes, and the release of digestive enzymes, causing autolytic soft tissue digestion [9]. The anoxic environment of the body increases the replication, transmigration, and activity of endogenous bacteria. This results in the bloating of soft tissues, which begins in the abdomen and extends to the rest of the body, increasing gas storage [10]. In addition, intrinsic physiological factors, such as disease, body mass index, etc., can influence the rate of decay and the metabolites produced [11,12,13]. Extrinsic biotic factors, such as external body microorganisms (mainly aerobes), insects, and micro- and macro-scavengers, coupled with extrinsic abiotic factors’ environmental parameters, e.g., humidity, temperature, sun exposure, ventilation, and burial context, affect the rate of decomposition [12,13,14,15].

Death is a multifaceted process in which each individual cell and tissue has its own metabolic homeostasis and time of functional cessation. The termination of functions of specific cells is a hallmark of death, defined by the dying process. On the other hand, some cells continue to exhibit vital signs, such as movement or growth and the ability to be cultured, after somatic death. Consequently, if cell death does not concurrently occur, the postmortem changes and the release of molecules fluctuates in terms of time, intensity, and profile. Advanced techniques based on the physiological omics approach have been developed in recent years; for example, miRNA, a noncoding RNA that changes as time since death passes, has been found in several tissues [16], and proteomic analysis has been applied to changes in human tissues that occur with death [17]. Additionally, untargeted metabolomics has enabled the identification of several metabolites that show potential as biomarkers [18], and recently, volatile organic compounds (VOCs) emitted by human body decay have been investigated [19].

A decaying body is physiologically associated with the release of different types of VOCs [20], which vary according to intrinsic and extrinsic factors, forming the volaboloma or odor mortis [21]. The volabolomic profile of decomposition is characteristic for each species and consists of different VOCs [22]. VOCs profiles are used to train cadaver dogs [23,24,25]. Consequently, released VOCs can be analyzed separately as individual compounds [19,20,21,22,23] or all together as a fingerprint, considering all the variable factors affecting the profile, as used in physiology [26,27,28,29,30,31]. In line with this approach, the creation of a Decompositional Odor Analysis (DOA) Database has begun to define the chemical fingerprints produced by VOCs during the decomposition process of human remains [32]. Recently, it was demonstrated that human postmortem myoblasts obtained from both ileopsoas and thyrohyoid muscles retain their ability to proliferate in vitro [33]. These myogenic populations, along with the fibers in which they originate, are interesting to investigate for forensic purposes, especially considering the extent of skeletal tissue, which accounts for approximately 45% of total body weight and covers almost all of the body’s surface, protected by skin. As a result, it can be reliable and accessible samples for PMI studies.

We studied the VOCs fingerprints emitted by human tissues and muscle cells with the aim to use them as complex biomarkers. VOCs fingerprints are formed by the combination of and interaction between multiple factors, which renders research regarding volatile profiles extraordinarily complex and difficult, so much so that the volabolomic approach to the open question of PMIs is required.

## 2. Materials and Methods

This study was carried out in agreement with the Declaration of Helsinki and approved by the local Ethics Committee (COET No. 6065-04.03.2021). Informed consents were signed by the deceased subjects’ family members.

Volabolomics is an omics science that investigates the volatile metabolites produced by the physiological activities of the body, tissues, and cells during the development, growth, aging, health, disease, and decay of the body.

The specimens used in this study were thyrohyoid and ileopsoas muscles collected from 40 (♂), 43 (♀), 45 (♂), and 71 (♀) year old corpses [for details regarding sampling and cell preparation methods, see Ref. [33]. Briefly, percutaneous needle biopsies consisted of the removal of tissue fragments from each muscle through a small accessory cutaneous cut. Sampling of the craniofacial muscle was carried out on the lateral margin, 2 cm from the clavicular insertion; removal of the ileopsoas muscle was carried out on the lateral margin 2 cm from its origin at the level of the greater trochanter. The collected tissues were divided into three samples for histopathological examination, tissue measurement, and preparation of dissociated cells. The control histopathological examination showed no signs of tissue injury.

Cells were isolated from muscle tissues by the explant procedure following the standard method previously described [33].

Volabolomic recordings, starting approximately twelve hours after death, were made using an e-nose sensor (iAQ-2000; Applied Sensor, CO2Meter.com, Beach, FL, USA) based on a metal oxide semiconductor (MOS) sensor, according to the standard analytical method [26,27,28,29,30,34]. This electronic nose automatically cleaned the sensor because of the temperature reached, which completely burned off the attached molecules. The e-nose was connected directly to the PC to record the signal at 1 Hz in real time and store it for analysis.

Skeletal fibers were collected in sterile tubes (15 mL), while myogenic cells were adherent to flasks (Corning, Glendale, CA, USA, 10 mL). The weights and numbers of cells were the same for all samples. Muscles were immediately analyzed. Cells were analyzed after their myogenic characterization [33].

Volabolomic recordings were performed in controlled conditions at 22 °C in an isolated biological hood to maintain sample sterility and avoid potential bacterial and mold effects on tissue or cell decay (Figure 1), in order to record “the pure” volabolomic signals from tissues and cells.

VOCs were continuously sampled, and later analyzed as a grand average of all samples at 0, 24, 48, and 72 h. The choice of these time intervals allowed the results to be double-checked with known methods.

Raw data were transformed and normalized using the formula (1/log_10_X)^y^; presented as means ± SD, and analyzed using MatLab software v. R2o23b, Jamovi 2.4.14, and Origin 2018b (9.55). Statistical analysis was performed with MANOVA and one-way post-hoc ANOVA tests, with the alpha level set at 0.01.

## 3. Results

VOCs were recorded by e-nose in real-time while the tissues and cells decomposed from 0 to 72 h. VOCs were compared as a grand average in specimen entire tissues vs. cells to investigate the general physiological behavior of decay. Further, VOCs’ frequencies were plotted for specimen entire tissues vs. cells. The PMI volabolomic fingerprint was illustrated from 0 to 72 h in respect to frequency and VOCs ppm.

### 3.1. Volatile Molecules Comparison

Volatile molecules emitted from cells and tissues were continuously sampled, resulting in their respective real-time volabolomic summaries, and later analyzed at 0, 24, 48, and 72 h. In Figure 2, statistical graphs depict VOCs emitted by cells and tissues over time, which show a common behavior of VOCs emission over the time course for both samples.

MANOVA analysis was applied to both specimens to ascertain differences as time passed between 0, 24, 48, and 72 h; in tissues return F_(3,882)_ = 108.06, *p* < 0.001, and in cells F_(3,326)_ = 8040.9, *p* < 0.001 (Table 1).

### 3.2. Volabolomic Profiles

Volabolomic profiles of cells and tissues at 0, 24, 48, and 72 h after death are displayed by VOCs frequency in Figure 3. VOCs’ distributions exhibited comparable behavior at almost all of the four intervals. Interestingly, from 48 to 72 h, both distributions became dispersed.

A post-hoc one-way ANOVA analysis applied to time intervals to determine the differences in tissues and cells over time returned significant differences (*p* < 0.001) in VOCs emitted during the tested time intervals (Table 2).

### 3.3. PMI Volabolomic Fingerprint

Finally, PMI volabolomic fingerprints could be returned as real-time plots of VOCs collected at 0, 24, 48, and 72 h after death. Volabolomic fingerprinting facilitated examination of increases in VOCs emission as time passed, which revealed different critical points at 24 h for whole tissues and at 48 h for isolated cells (Figure 4). An additional consideration was that the variability in VOCs emission increased dramatically at 72 h for tissues (Figure 4).

## 4. Discussion

The PMI plays a central role in forensic pathology [21,32,35,36]; establishing the PMI is one of the most challenging tasks in criminal investigations. In fact, current methods for PMI determination exceed 40% error, even when combining conventional methods, due to the complexity of putrefaction mechanisms [37]. Therefore, a quantitative, objective, and reliable method to estimate PMI is crucial. Recently, a promising method has been provided by omics science approaches, for example, metabolomics. Metabolomics can provide information regarding the presence and relative abundance of a wide range of small molecules and changes in the levels of amino acids, oligopeptides, lactate, alanine, pyruvate, glucose, and fatty acids in tissues (liver, kidney, spleen, brain, heart, and muscle tissues) [38,39,40]. In particular, lactate and pyruvate levels in muscle tissue have been correlated with the PMI. This evidence suggests that one key to studying PMIs is the physiological approach [41].

Metabolomics also includes volatile metabolites, VOCs, which have been particularly studied in the volabolomics field. VOCs are the subject of increasing interest in the study of PMIs, and are the target of the DOA database [32]. It is interesting to note that the best-known molecules characterizing cadaveric effluvium, cadaverine, putrescine, and mercaptans, return a generic putrefactive scent and do not contribute to the characteristic species-specific volaboloma mortis, which is instead a vital clue associated with micro- and macro-scavengers and insects [21,32,42]. In fact, high variability among mammals has been demonstrated in the volatile chemical classes of aldehydes, amines, alcohols, and ketones, which characterize the volabolomas of human, dog, pig, and deer carcasses [32].

In addition, the “odor”, more properly called the volabolome, of the living body is a biometric characteristic unique to all, and can be used for individual identification, even of monozygotic twins [43,44]. In fact, both qualitative and quantitative volabolome differences have been observed between human sexes, dependent on specific compounds, primarily esters and aldehydes [44,45,46]. In addition, common species compounds are present in different ratios between the sexes, indicating qualitative similarities between individuals with quantitative differences [44,45,46].

As a result, human remains release VOCs that are different from those of other mammals, which characterizes the human VOCs fingerprint [47,48]. Moreover, within the same species, there are individual differences that constitute the personal chemical signature used by so-called “molecular” dogs for forensic applications. This individual signature probably even persists after death for some time [44,49]. The application of the e-nose for recording and measuring human VOCs has enabled the identification and classification of the volabolome of our species [50]. Recently, the e-nose has been used for weeks to record changes in volatile metabolites released from a corpse laid on the surface during the stages of decomposition, demonstrating its usefulness as a detection tool [51].

However, there are no unified recommendations for conducting forensic analysis based on the volabolome fingerprinting of cadavers, primarily because of the mapping methods, analytical approaches, and signature investigations used.

In this study, we exclusively investigated the use of volabolomics, metabolomics investigating volatile metabolites, using a physiological approach because of the possibility of fingerprinting cadaveric VOCs at different times. For this preliminary experimental phase, we used tissues and muscle cells, in accordance with previous studies, for up to 72 h [38].

The method applied in this study mirrored that used in standard physiological investigations [26,27,28,29,30,31,34]. In these physiological studies, the volabolomic fingerprint itself represents a complex and specific “biomarker” applied to the study of a given pathology or condition [26,27,28,29,30,31,34]. The omics approach, while producing a large data set, is useful in simplifying the results if one thinks in terms of a multi-dimensional fingerprint. In summary, the volabolomic approach returns a picture of metabolism under conditions of health and disease, that is, under physiological homeostasis or its alteration [26,27,28,29,30,31,34]. Similarly, during corpse decomposition, metabolism remains a physiological phase and generates a specific set of molecules that depend on intrinsic and extrinsic parameters that should be precisely studied in its complexity to try to simplify it for the purposes of forensic medicine.

The first result is that both tissue and muscle cells emit putrefaction VOCs recordable by the e-nose, which is in agreement with previous studies [50,51]. Moreover, VOCs emitted from both cellular and tissue samples showed homologous putrefaction behavior. In addition, this pattern was in line with results obtained in the metabolomic study of vitreous humor decomposition, which changes gradually in quantitative terms with increasing PMIs [52]. Another result of this study is the effectiveness of sampling superficial muscle tissue; it is very easy to sample, has less emotional impact than sampling the eye does for close relatives; and is more stable post-mortem than other organs or body fluids are. These are all advantages in superficial muscle tissue’s use as a tissue marker for PMI investigations.

The volabolomic profiles obtained in this study from tissues and cells agree with those of the previous study [52], particularly regarding qualitative changes with increasing PMIs. Accordingly, as decay increases, liquid and volatile metabolites progressively change both qualitatively and quantitatively.

In this study, we also showed that volatile compounds progressively increase with increasing PMIs. This phenomenon is likely related to the breakdown of molecules into smaller and more volatile elements. The changes leading to the volatile metabolic portion appeared homogeneous in both tissues and cells. In agreement with the literature, we used muscle VOC profiles to reduce individual variability [53]. Differences between cadaver organs and tissues may be due to various reasons, ranging from cause of death, diet, lifestyle, time since death, etc. [53].

This approach obviously suffered from several limitations, such as the number of repetitions, lack of reproducibility among samples, and diversity among cadavers. In addition, longitudinal monitoring from the original cadaver state was not possible, which means that understanding the evolution of these products over time is challenging.

However, the proposed volabolomics method seems important regarding its potentially significantly contribution to the systematic investigation of the time-dependent emission of VOCs from cadavers under controlled and standardized conditions. It is a promising tool for PMI estimation because VOCs emitted during the decay of body tissue and cellular changes can be measured in real time based on time intervals of 0, 24, 48 and 72 h, and compared. This is in agreement with metabolomics, which returns a model with high predictive ability [40,52].

Although prediction errors increase with increasing PMIs if <24, 24–48, or >48 h [52], as intrinsic and extrinsic inferences can cause a progressive increase in biological complexity [52], the omics approach might be the best approach to reduce this complexity with the development of fingerprinting as a complex “biomarker,” as we suggest in this paper.

Finally, this paper suggests for the first time in the literature, to the best of our knowledge, that a human skeletal muscle trend can be associated with PMIs, as can a myogenic stem cells trend. Furthermore, we point out that the choice of sample muscles was due to the fact that the thylakoid is superficial while the ileopsoas is deep and large. Consequently, the thylakoid was chosen because it is easily accessible; the ileopsoas was chosen because, given its size, it should always be present, even under advanced scrapping conditions. We can say that VOCs fingerprinting the major cellular components of skeletal muscle makes the search for volatile profiles, in other words, skeletal muscle-dependent volaboloma mortis, extraordinarily interesting and perhaps promising for SME identification. Further studies are required to definitively establish the volaboloma mortis bank for PMIs and associated variables based on changing conditions intrinsic and extrinsic to the cadaver. For this purpose, the e-nose is the most suitable device for VOCs collection in the field and/or autopsy room. In addition, the e-nose we used is highly moisture stable and can be used in real time for days, weeks, or months.

In addition, broader VOCs fingerprinting applications may be found, for example, in dog training procedures for detecting human remains, and in the study of insect attractants and scavengers. Future studies could develop mathematical models that also use artificial intelligence to investigate variables. Volabolomic fingerprinting is an objective system that can be extended to all possible environmental variables and provides an accurate error range.

## Figures and Tables

**Figure 1 biomolecules-14-00286-f001:**
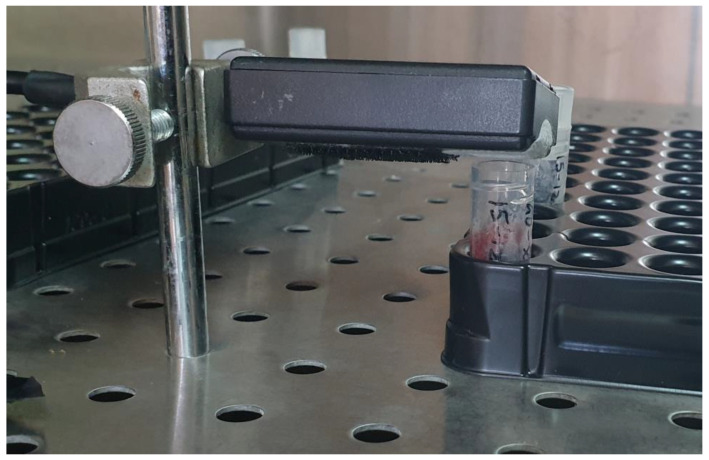
Recording system. Set up for real-time recording of volatile molecules emitted immediately after excision (that we assumed as 0 h) and 24, 48, and 72 h following muscle excision.

**Figure 2 biomolecules-14-00286-f002:**
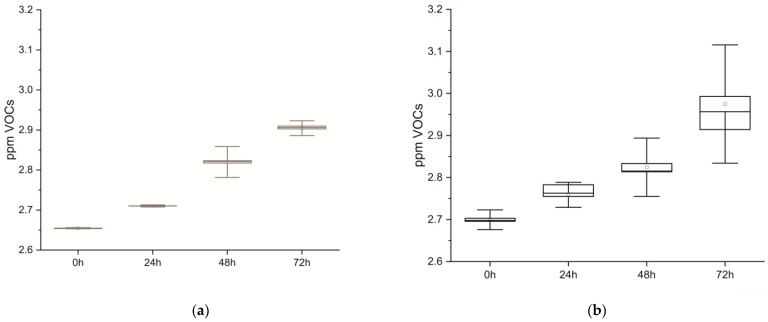
Statistical box and whiskers plot. Real-time recordings of volatile molecules (VOCs) emitted at 0, 24, 48, and 72 h after death: (**a**) muscle cells and (**b**) tissues.

**Figure 3 biomolecules-14-00286-f003:**
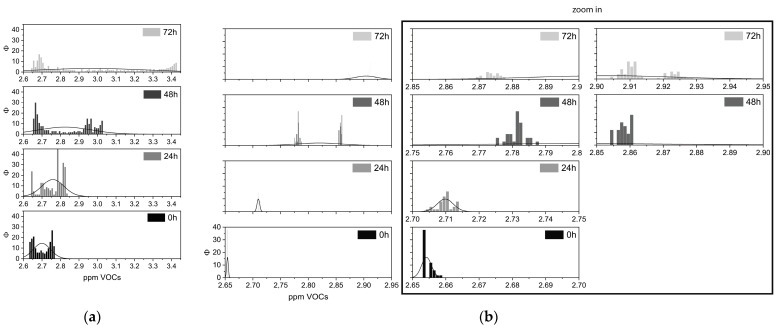
Frequency VOCs plots. Real-time recordings of volatile molecules emitted at 0, 24, 48, and 72 h after death, fitted by relative normal curve, from (**a**) muscle tissues and (**b**) cells; the inset shows a zoom in, in order to better visualize the histograms, and the *X*-axis has been amplified.

**Figure 4 biomolecules-14-00286-f004:**
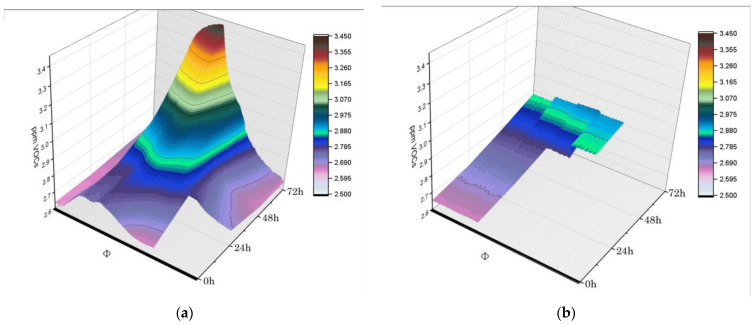
PMI volabolomic fingerprints. Real-time VOCs were plotted per specimen, (**a**) tissues vs. (**b**) muscle cells, at 0, 24, 48, and 72 h after death.

**Table 1 biomolecules-14-00286-t001:** Statistical mean and standard deviation (SD) of volabolomic profiles at 0, 24, 48, and 72 h, emitted from muscle cells and tissues.

Time since Death	0 h	24 h	48 h	72 h
ppm VOCs tissues	2.70 ± 0.05	2.76 ± 0.06	2.82 ± 0.14	2.98 ± 0.28
ppm VOCs cells	2.65 ± 0.002	2.71 ± 0.002	2.82 ± 0.002	2.90 ± 0.019

**Table 2 biomolecules-14-00286-t002:** Statistical one-way ANOVA analysis of volabolomic profiles; at 0, 24, 48, and 72 h, volatile molecules emitted from collected muscle cells and tissues continuously expressed ppm VOCs.

Time since Death	0–24 h	0–48 h	0–72 h	24–48 h	24–72 h	48–72 h
Tissues	F_(1,406)_ = 117.4	F_(1,406)_ = 126.4	F_(1,406)_ = 158.4	F_(1,476)_ = 44.9	F_(1,476)_ = 134.6	F_(1,476)_ = 54.8
Cells	F_(1,118)_ = 22,676	F_(1,148)_ = 369,748	F_(1,178)_ = 10,776	F_(1,148)_ = 104,602	F_(1,178)_ = 6477	F_(1,208)_ = 1836

## Data Availability

Data is unavailable due to ethical restrictions.

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
