# Peer review of "Volabolomic Fingerprinting for Post-Mortem Interval Estimation: A Novel Physiological Approach"

_biomolecules, 2024, doi:10.3390/biom14030286_

Round 1

Reviewer 1 Report

Comments and Suggestions for Authors

Line 17 – define VOC the first time you use an abbreviation

Line 22 – Is there a more formal term to use then “electronic nose” to describe the technology?

Line 33 – is a pivotal _____, what? There appears to be a word missing here.

Line 72 – Can remove “a” from “a human body”

Line 102 – “for details see 34” What does this mean? See reference 34? Is this for more details on the cadavers or on the procedure? Please clarify.

Line 138 – change “during tissues and cells decay” to “while the tissues and cells decomposed”

Line 141 – change “showed” to “displayed”  

General -- The “Discussion” section appears to be written by a different person then the rest of the document. The English language is clear and does not require any edits, whereas the English language quality is lower in the rest of the document. I would recommend the co-author who wrote the “Discussion” section read and make edits to the English translations in the rest of the document.

Comments on the Quality of English Language

he “Discussion” section appears to be written by a different person then the rest of the document. The English language is clear and does not require any edits, whereas the English language quality is lower in the rest of the document. I would recommend the co-author who wrote the “Discussion” section read and make edits to the English translations in the rest of the document.

Author Response

First of all I would like to thank the referee for his work in improving the paper

Ref #1

Authors reply:

  • Line 17 – define VOC the first time you use an abbreviation. Ok, done.
  • Line 22 – Is there a more formal term to use then “electronic nose” to describe the technology? No, but we modify in “electronic nose sensor device”.
  • Line 33 – is a pivotal _____, what? There appears to be a word missing here. Right! miss “element”.
  • Line 72 – Can remove “a” from “a human body”. Ok, done.
  • Line 102 – “for details see 34” What does this mean? See reference 34? Is this for more details on the cadavers or on the procedure? Please clarify. According referee we modify in “For details on sampling and cell preparation methods, see Ref 34”.
  • Line 138 – change “during tissues and cells decay” to “while the tissues and cells decomposed”. Ok, thanks we use the suggestion.
  • Line 141 – change “showed” to “displayed”. Ok, thanks we modify according to the Referee.
  • General -- The “Discussion” section appears to be written by a different person then the rest of the document. The English language is clear and does not require any edits, whereas the English language quality is lower in the rest of the document. I would recommend the co-author who wrote the “Discussion” section read and make edits to the English translations in the rest of the document. In agreement with the Referee the discussion has been re-edit.

Reviewer 2 Report

Comments and Suggestions for Authors

This article presents an interesting investigation of volatiles emitted from tissue and muscle cells using an electronic nose. This a novel area that is currently gaining increasing attention. The work is interesting and with some modifications and added information will be of value. 

I have some overall comments and questions around this work. In this study the time-periods used are very early where PMI is not challenging to ascertain using other methods. Was the technology tested with later time-periods? If not, then the need for this device and the added value should be further discussed.

One major concern is around the sample types. The study investigated volatiles emitted from thyrohyoid and ileopsoas muscles, how can this data be extrapolated to whole human victims which is likely to be what is encountered in the short period of time investigated in the current study?

Which MOS sensors were used and on what basis were these selected? How were the sensor responses treated if at all prior to analysis and what is the feature used to generate the data used herein? The paper also reports ppm values of VOCs, how was this determined? Were tests done using analytical standards to confirm these concentrations? If not the ppm cannot be known.

Was chemical characterisation carried out to confirm the responses by the electronic nose? Which volatiles were responsible for the e-nose signal?

Cells and tissues were investigated from donors, were there differences observed between donors? It is not clear in all instances which data is used and if the data for all donors at the same timepoints were averaged or otherwise treated.  

Please see below for specific comments:

Line 101: How soon after death were the donors and subsequently the tissues obtained?

Line 131: “VOCs were sampled continuously and later analyzed at 0, 24, 48 and 72 hours.” – does the e-nose have a cleaning stage at any point or was this eliminated? Was gas or temperature used to remove analytes from the MOS sensors at any point during this time?

Line 141: “Finally, PMI Volabolomic fingerprint was showed.” It is unclear what is meant by this.

Figure 2: which sensors was this based on? Were all sensors responding or is this depicting selected ones? Missing legend to explain the differing plots per time event. Was this data from cells or tissues and all donors?

Figure 3: The x-axis is unclear, how was the concentration determined and what are the differing columns representing? Are these different sensors?

Line 232: “Muscle tissue has rarely been the subject of PMI research.” – there are a number of existing works investigating biomarkers from muscle tissue for PMI using proteomics, metabolomics, lipidomics and volatilomics so I would revise this statement.

Author Response

First of all I would like to thank the referee for his work in improving the paper

Ref #2

Authors reply:

  • In this study the time-periods used are very early where PMI is not challenging to ascertain using other methods. Was the technology tested with later time-periods? If not, then the need for this device and the added value should be further discussed. In agreement with the Referee, in this first study we use a time period in which our results are easily controlled by other methods, we will perform further experiments over a longer period; to be clear, we have added one sentence in mat and met in line 131. “The choice of this time period allows the results to be double-checked with known methods.” and one in discussion line 270. “For this purpose, the e-nose is the most suitable device for collecting VOCs in the field and/or in the autopsy room. In addition, the e-nose used is highly moisture stable and can be used in real time for days, weeks or months.”
  • One major concern is around the sample types. The study investigated volatiles emitted from thyrohyoid and ileopsoas muscles, how can this data be extrapolated to whole human victims which is likely to be what is encountered in the short period of time investigated in the current study? According the referee observation to better explain we also add a sentence in discussion line 264. “The choice of muscles is due to thyrohyoid is superficial while the ileopsoas is deep and large. Consequently, the former was chosen because it is easily accessible; the latter because given its size it should always be present even in advanced scrapping conditions.”
  • Which MOS sensors were used and on what basis were these selected? All info about the sensor is already in mat and met line 112. “e-nose sensor (iAQ-2000; Applied Sensor), based on a metal oxide semiconductor (MOS) sensor” the choice of this e-nose is due to its high humidity stability, the capacity to continuously record in real time for months.
  • How were the sensor responses treated if at all prior to analysis and what is the feature used to generate the data used herein? The only data manipulation is as explained in mat and met line 133. “Raw data were transformed and normalized using the formula (1/log10X)y”.
  • The paper also reports ppm values of VOCs, how was this determined? Were tests done using analytical standards to confirm these concentrations? If not the ppm cannot be known. The ppm is returned by the instrument directly, however in a previous paper Mazzatenta A, Pokorski M, Cozzutto S, Barbieri P, Veratti V, Di Giulio C. Non-invasive assessment of exhaled breath pattern in patients with multiple chemical sensibility disorder. Adv Exp Med Biol. 2013;756:179-88. doi: 10.1007/978-94-007-4549-0_23 used a standard analytical instrument, Wohler A600 gas analyzer (Wohler USA Inc. Danvers, MA), to verify the O2 and CO2 measurements done by the e-nose iAQ-2000.
  • Was chemical characterisation carried out to confirm the responses by the electronic nose? Which volatiles were responsible for the e-nose signal? Referee's question is in progress in a parallel study using a standard analytical chemistry investigation. In this study, we used an adsorbent material (membrane with different activations) to collect a broad spectrum of VOCs and then characterize it in an analytical chemistry laboratory. However, the e-nose fingerprint might be more informative than the simple list of chemicals, especially since the e-nose reacts to the entire bouquet of VOCs emanating from cadaveric tissues.
  • Cells and tissues were investigated from donors, were there differences observed between donors? It is not clear in all instances which data is used and if the data for all donors at the same timepoints were averaged or otherwise treated. Right! we agree with the Referee and explain that we used averaged data of all subjects, we do not observe important differences between donors, consequently we add the following sentence in Mat and Met “VOCs were sampled continuously and later analyzed, as grand average of all samples, at 0, 24, 48, and 72 hours” and in results we add “VOCs were compared as grand average in both specimen entire tissues vs. cells to in-vestigate the general physiological behavior of decay” line 143.

Please see below for specific comments:

  • Line 101: How soon after death were the donors and subsequently the tissues obtained? The recording start rapidly in about 12 hours after death, we add a sentence in line 112 “Volabolomic recordings, starting about twelve hours after death”.
  • Line 131: “VOCs were sampled continuously and later analyzed at 0, 24, 48 and 72 hours.” – does the e-nose have a cleaning stage at any point or was this eliminated? NO, this device do not have any cleaning stage because the sensor burn completely and continuously the molecules. Was gas or temperature used to remove analytes from the MOS sensors at any point during this time? Right the MOS is clean by itself with Temperature. We add the following sentence “The electronic nose used automatically cleans up the sensor because of the temperature reached, which completely burns off the attached molecules.” in mat and met at line 114.
  • Line 141: “Finally, PMI Volabolomic fingerprint was showed.” It is unclear what is meant by this. Ok, we explain better modifying the sentence as follow “The PMI Volabolomic fingerprint was illustrated from 0 to 72 h in respect to frequency and VOCs ppm”.
  • Figure 2: which sensors was this based on? Were all sensors responding or is this depicting selected ones? Missing legend to explain the differing plots per time event. The figure has been reformatted completely. Note, the sensor is always the same. Was this data from cells or tissues and all donors? This point has been clear in mat and met line 131.
  • Figure 3: The x-axis is unclear, how was the concentration determined and what are the differing columns representing? Are these different sensors? The figure has been completely reformatted in order to be more clear. Note, the sensor is always the same.
  • Line 232: “Muscle tissue has rarely been the subject of PMI research.” – there are a number of existing works investigating biomarkers from muscle tissue for PMI using proteomics, metabolomics, lipidomics and volatilomics so I would revise this statement. Ok was removed.

Reviewer 3 Report

Comments and Suggestions for Authors

General comments:

This paper (biomolecules-2858669) reported an experimental study on VOCs fingerprint emitted by human tissues and muscle cells for estimating post-mortem interval. The article is well-formed, and the results of this study are novel and meaningful. However, the quality of the charts needs to be improved. In general, I think that after a minor revision, the article can be accepted for publication.

Specific Comments:

Lines 22,131, 144, 157, 173, and 180: There should be a “,” before “and”.

Line 117: There should be a space between “10” and “ml”.

Figures 2 and 3: The pictures can be colored to show them more clearly. There should be spaces between numbers and their units.

Figure 2: Please add legend in the Figure. What do the dark and light boxes represent respectively?

Tables 1 and 2: Please format the table because the first line is empty.

Please check and uniform the reference format.

Author Response

First of all I would like to thank the referee for his work in improving the paper

Ref #3

Authors reply:

However, the quality of the charts needs to be improved. In general, I think that after a minor revision, the article can be accepted for publication.

Specific Comments:

  • Lines 22,131, 144, 157, 173, and 180: There should be a “,” before “and”. Ok done.
  • Line 117: There should be a space between “10” and “ml”. Ok done.
  • Figure 2: Please add legend in the Figure. What do the dark and light boxes represent respectively? Figures 2 and 3: The pictures can be colored to show them more clearly. There should be spaces between numbers and their units. In agreement with referee suggestion Fig.2 and Fig.3 has been completely reformat.
  • Tables 1 and 2: Please format the table because the first line is empty. Ok done.
  • Please check and uniform the reference format. Ok done.

Round 2

Reviewer 2 Report

Comments and Suggestions for Authors

My comments have been addressed.